# Improving Recognition of Treatable Rare Neuromuscular Disorders in Primary Care: A Pilot Feasibility Study

**DOI:** 10.3390/children9071063

**Published:** 2022-07-17

**Authors:** Federica S. Ricci, Rossella D’Alessandro, Martina Vacchetti, Anna Salvalaggio, Alessandra Somà, Giorgia Daffunchio, Marco Spada, Renato Turra, Marisa Bobbio, Alessandro Ciuti, Chiara Davico, Benedetto Vitiello, Tiziana E. Mongini

**Affiliations:** 1Division of Child and Adolescent Neuropsychiatry, Department of Public Health and Pediatric Sciences, University of Turin, 10126 Turin, Italy; rossella.dalessandro@unito.it (R.D.); mvacchetti@cittadellasalute.to.it (M.V.); anna.salvalaggio@unito.it (A.S.); alessandra.soma@unito.it (A.S.); giorgia.daffunchio@edu.unito.it (G.D.); chiara.davico@unito.it (C.D.); benedetto.vitiello@unito.it (B.V.); 2Division of Pediatrics, Department of Public Health and Pediatric Sciences, University of Turin, 10126 Turin, Italy; marco.spada@unito.it; 3Division of Primary Care Pediatrics, Health District ASL TO4, Via Po 11, 10034 Turin, Italy; renatoturra@libero.it; 4Division of Primary Care Pediatrics, Health District ASL TO5, Piazza Silvio Pellico 1, 10023 Turin, Italy; bobbiomarisa@gmail.com; 5Division of Child and Adolescent Neuropsychiatry, Health District ASL TO5, Piazza Silvio Pellico 1, 10023 Turin, Italy; ciuti.alessandro@aslto5.piemonte.it; 6Division of Neurology 1, Department of Neuroscience, University of Turin, 10126 Turin, Italy; tizianaenrica.mongini@unito.it

**Keywords:** neuromuscular disorders, disease modifying treatments, early diagnosis, motor development milestones, primary care pediatricians, screening, public health

## Abstract

Innovative targeted treatments for neuromuscular disorders (NMDs) can dramatically improve the course of illness. Diagnostic delay, however, is a major impediment. Here, we present a pilot project aimed at assessing the feasibility of a screening program to identify children at high risk for NMDs within the first 30 months of life. The Promoting Early Diagnosis for Neuromuscular Disorders (PEDINE) project implemented a three-step sequential screening in an area of about 300,000 people with (1) an assessment of the motor development milestones to identify “red flags” for NMDs by primary care pediatricians (PCPs) as part of the routine Health Status Check visits; (2) for the children who screened positive, a community neuropsychiatric assessment, with further referral of suspected NMD cases to (3) a hospital-based specialized tertiary care center. In the first-year feasibility study, a total of 10,032 PCP visits were conducted, and twenty children (0.2% of the total Health Status Check visits) screened positive and were referred to the community neuropsychiatrist. Of these, four had elevated creatine kinase (CK) serum levels. This pilot study shows that screening for NMDs in primary care settings is feasible and allows children at high risk for muscular disorder to be promptly identified.

## 1. Introduction

Neuromuscular diseases (NMDs) represent a heterogenous group of disorders impairing the function of muscles, motor neurons, peripheral nerves, and/or neuromuscular junctions, with a very broad phenotypic spectrum. To date, more than 1100 monogenic NMDs, linked to 641 different genes, have been identified in humans [1]. The development of disease-modifying treatments (DMTs) for NMDs has been the major advance in the last ten years, leading to the approval of transformative treatments with dramatic effects on genetic diseases such as congenital myasthenia syndrome (CMS), spinal muscular atrophy (SMA), and Pompe disease (PD) [2]. Promising preliminary data have also been reported in limb girdle muscular dystrophy, X-linked myotubular myopathy, and in Duchenne muscular dystrophy (DMD), for which five drugs have so far reached regulatory approval [2,3]. The clinical landscape for DMTs is widening, with dozens of new treatments on the horizon. For example, a recent systematic review of clinical trials of adeno-associated virus (AAV)-based gene therapies identified 149 unique clinical trials [4]. The number of clinical trials of AAV-based gene therapies initiated annually increased from less than 5 in 2007–2010 to 17 for CNS disorders and 15 for musculoskeletal disorders in 2015–2019 [4].

Over the last decades, the multidisciplinary care of NMDs has also evolved, with the focus now placed on the early diagnosis and anticipatory treatment strategies addressing expected and potentially modifiable disease complications [5,6,7,8,9]. Diagnostic delay in NMDs, however, is a significant hurdle as the irreversible muscle damage that occurs prior to the diagnosis can significantly limit treatment efficacy [10,11,12,13,14]. Even in the absence of a rapid muscle destruction process, the long diagnostic journey can cause decades of limitation in the quality of life before the NMD is diagnosed and the appropriate treatment prescribed [3].

To foster a prompt diagnosis of NMDs, three methods have been considered: newborn screening (NBS), pediatric population screening, and screening of a population at risk. NBS programs have the potential to identify individuals with rare treatable conditions at an early stage and to shorten the time to starting disease-changing therapies, thus improving health, development, and life expectancy [15]. They are considered an important measure of secondary disease prevention and one of the greatest advances of modern public health with significant individual, socio-economic, and societal benefits [16]. Starting more than 50 years ago, with phenylketonuria as the first target disease [17], technical advances, such as tandem mass spectrometry [18] and genetic screening tests [19], and the availability of DMTs, led to a continuous extension of NBS programs. The ten modified criteria proposed by Wilson and Jungner [20] are broadly used worldwide to determine whether screening for a disease should be included in an NBS panel, still at national policy level there is a highly variable composition of NBS disease panels [20].

A recent review identified seven diseases with a clear neuromuscular involvement that have been targeted by NBS programs over the last twenty years: SMA, PD, DMD, myotonic dystrophy type 1 (MD1), Krabbe disease, X-linked adrenoleukodystrophy (X-ALD), and metachromatic leukodystrophy (MLD) [3]. SMA is certainly the disease for which NBS has the greatest consensus, given the demonstrated importance of early intervention and the dramatic efficacy of pre-symptomatic treatment [21,22]. NBS of PD brings different concerns, the most significant being the proportion at birth of late onset forms for which there is today no indication of early treatment, compared to the less frequent infantile form, for which the NBS could allow an early treatment initiation [3]. As for the other neuromuscular diseases with metabolic origin (i.e., Krabbe disease, X-ALD, and MLD), the setup of specific and laborious assays tends to hamper the implementation of corresponding NBS [3]. Regarding DMD, only about 30% of patients could potentially benefit from early detection, with steroids usually not being prescribed before the age of three years, and all approved treatments being mutation specific. However, apart from the aforementioned seven screenable diseases, most NMDs have neither a specific biomarker nor a prevalent molecular defect so that screening for these disorders is not possible. A clear example can be found in congenital myasthenic syndromes (CMS): low-cost treatments such as salbutamol or pyridostigmine can avoid sudden death or disability in the course of a long diagnostic journey, but to date, 34 genes have been described to be involved in CMS, hampering the screening of the disease [3]. Identification of such disorders at birth should therefore be carried out only after a technological paradigm shift, moving towards expanded genetic-based approaches (whole exome or targeted exome sequencing), which probably will become cheaper and available in the course of the next few decades. However, while sequencing of a specific genomic region, whole exome, and genome sequencing to assist in the diagnosis of symptomatic newborns pose few concerns, whole-exome and genome sequencing as screening tools for all newborns can raise many scientific and ethical concerns [23].

Regarding pediatric population-based screening, the few attempts for NMDs were performed using serum CK levels at different ages and in different clinical settings, but the lack of specificity of CK makes the use of this test difficult to interpret when it is used at the population level [24].

The term ‘screening of a population at risk’ means the application of expanded metabolic or genetic tests to a group of individuals with risk factors for NMDs, according to their medical history or selected clinical signs. The key role of the general practitioner has already been highlighted for screening adult patients [25]. However, diagnostic support for NMDs is needed, due to a lack of experience of primary care physicians with NMDs. New systems to remind primary care physicians of the risk for rare diseases are highly desirable, as underscored by multiple reports addressing the problem of diagnostic delay in different disease groups. In 2016, Grigull et al. proposed a proof-of-concept trial of the administration of a 46-item questionnaire to selected patients regarding six NMDs, analyzed with a fusion algorithm [26]. The sensitivity of the system was 93–97% for individuals with PD, with MD1/myotonia and for individuals without NMD, but only 69% in SMA and 81% in amyotrophic lateral sclerosis (ALS) patients. In the prospective trial, 89% of the diagnoses were correctly predicted by the computerized system, but additional studies are required to measure the effectiveness of this approach in clinical settings [26]. Regarding DMD, checking CK early in the evaluation of boys with unexplained developmental delay was proposed more than ten years ago [10]. Recently, Van Dommelen et al. analyzed a series of DMD patients focusing on early development milestones and found delays in the attainment of motor and non-motor milestones within the first few months of life, suggesting that this approach could aid in the earlier diagnosis of DMD [27]. In the UK, some authors have already proposed a specific screening for DMD at the 27–30-month health check with CK testing and the aid of a DMD mnemonic MUSCLE acronym (Motor milestone delay, Unusual gait, Speech delay, CK as soon as possible, Lead to, Early diagnosis of DMD) [28], but no real-world clinical experience has been reported so far.

Here, we report on a pilot study aimed at promoting the early diagnosis of NMDs in children within the first two years of life. The study was conducted in a health district of about 300,000 inhabitants in the metropolitan area of Turin (Italy), with the aim of examining the feasibility of a collaborative care three-step screening approach involving, in sequence, primary care pediatricians (PCP), community child neuropsychiatric services, and a hospital-based specialized center in NMDs. In particular, the study was meant to address the following questions: (a) what is the acceptability by PCPs of conducting a brief screening of motor developmental milestones during the routine well-child care visits of the national health systems during the first two years of life? (b) what is the yield of PCP-identified “red flags”? (c) what is the follow-up of the children who screened positive regarding neuropsychiatric visits and CK serum levels?

## 2. Materials and Methods

### 2.1. Program Development

PEDINE (Promoting Early DIagnosis in NEuromuscular diseases) is an observational prospective study aimed at creating a network between the community of PCPs and eight principal Child Neuropsychiatry Units in the Piemonte Region (Italy), under the coordination of the Neuromuscular Regional Referral Center (University Hospital ‘Città della Salute e della Scienza di Torino’, Associated National Center in the European Reference Network ERN EURO-NMD). The primary objective was to identify and promptly assess suspected cases of motor milestone delay as a way of reducing the time to diagnosis of DMD, SMA, PD, and other NMDs. The secondary objective was to develop and apply an organizational “hub and spoke” model for infants and children with developmental delay.

PEDINE implemented a 3-step sequential screening with (1) a universal systematic assessment of the critical milestones of neuro-motor development to identify “red flags” for NMDs in the first two years of life by PCPs as part of the routine Health Status Check of the National Health Service; (2) for the children who screened positive, a neuropsychiatric assessment by community neuropsychiatrists, who referred suspected NMD cases to (3) a hospital-based specialized tertiary care center. A dedicated email address for reporting children with ‘red flags’ of suspected muscle impairment was also established.

In 2018, a preparatory design work was conducted by a small group of 30 selected PCPs belonging to the Italian Federation of Pediatricians (FIMP) in collaboration with the Neuromuscular Regional Referral Center of Turin. In Italy, each PCP cares for up to 800 children, including about 200–300 under 4 years of age [29]. Each child undergoes periodic Health Status Check visits as per the National Health System policies, with referral to local specialists in case of abnormal findings. In the first three years of life the scheduled visits are at 15 days, 2–3 months, 4–5 months, 8 months, 10–11 months, 15–18 months, and 24–30 months of age. The working group identified for each scheduled visit a specific ‘red flag’ for a missed motor milestone (Table 1). A positive ‘red flag’ would prompt quick referral to the Neuromuscular Regional Referral Center of Turin through a dedicated communication channel. To facilitate the assessments, a summary table was created with illustrations of ‘red flags’.

In total, 19 out of 30 PCPs actively participated in the three-month preliminary project (from October to December 2018), performing 1871 scheduled visits and referring 13 children. Due to the low numbers, as expected, no NMD case was detected. However, one child was diagnosed with a non-NMD rare genetic disease, and another with a neurodevelopmental disorder, and both children could be referred to dedicated specialist services for prompt treatment. The study showed the feasibility of using a ‘red flag’ system within the routine Health Check Visits, the utility of a dedicated communication channel between PCP and specialist neuropsychiatrist, and the complexity of differential diagnosis. According to these results, the percentage of referral for motor milestone delay was 0.7% (CI 95% 0.3–1.0) of the total Health Status Check visits. If calculated on 25,937 newborns per year (2020 newborns in the Piemonte Region, Italy) [30], 544,677 visits will be performed by pediatricians in the first 36 months of life, with 3813 referrals expected.

According to these data, from 2020, eight Child Neuropsychiatric Centers of the Regional Districts, covering the whole area of the Piemonte Region, and PCPs of the National Health System of the Piemonte Region were involved, with a “hub and spoke” model. The ‘red flag’ system, already tested in the preliminary project, was used. In the presence of a positive ‘red flag’, the child was to be referred through a dedicated communication channel to the competent district Child Neuropsychiatry Unit, with appointment within 3–10 days (depending on the type of red flag and therefore the type of delay). An alerted child neuropsychiatrist would then proceed with the differential diagnostic protocol for motor delay, with the supervision of the Neuromuscular Regional Referral Center. The diagnostic protocol included the serum CK level, the alfa-glucosidase dosage on Dried Blood Spot (DBS), and the genetic test for SMA (MLPA, Multiplex ligation-dependent probe amplification) for each reported child, with more complete neuromuscular diagnostic tests at the Neuromuscular Regional Referral Center only for children with specific neuromuscular signs (See Appendix A for the complete diagnostic protocol). Each district required different timelines, also considering the COVID-19 pandemic.

### 2.2. Program Implementation

This pilot project was conducted in a district of the metropolitan area of Turin (ASL TO5, with about 300,000 inhabitants) from January to December 2021. A total of 32 PCPs, 10 community child neuropsychiatrist, and one regional hospital-based specialized pediatric NMD center participated in the study.

The project included a first informative section and an operative part. For the first informative section, a formal educational meeting was organized for the PCPs and the child neuropsychiatrists of the ASL TO5 District, utilizing slides, video, and printed material specifically aimed at the normal motor development in healthy children and the principal neurological and neuromuscular disorders. For the second operative part, a dedicated email address was created by the district Child Neuropsychiatry Unit, with a guaranteed specialistic visit within 3–10 days, depending on the age of the patient and the ‘red flag’ identified. The Neuromuscular Regional Referral Center helped to coordinate the informative flux and to complete more complex cases.

### 2.3. Data Analyses

Descriptive statistics were applied to the data.

## 3. Results

In the 12-month study period, a total of 10,032 PCP visits were conducted (Table 2).

Twenty children (8 males, 12 females) were flagged by the PCP for deficit in reaching one or more motor development milestones and were therefore referred to the community child neuropsychiatrist (Figure 1). They represent the 0.2% of the total Health Status Check visits (CI 95% 0.1–0.3). In total, 4 out of these 20 patients were referred from the district child neuropsychiatrist to the Neuromuscular Regional Referral Center for clinical signs consistent with a neuromuscular disease.

The main ‘red flags’ that were identified were: 6–9 months (sitting position) and 10–11 months (shuffling or crawling), with 8 and 6 patients, respectively (see Table 3 for details). It should be noted that 5 out of 20 children were reported in the first months of life, notoriously the most significant for identifying the most severe phenotypes.

Of the 16 children assessed at the district level, 5 did not attend the specialistic evaluation (drop out), 10 had normal CK level but did not attend further examinations (diagnostic tests for Pompe disease and SMA), and 2 had mild hyperCkemia (<300 UI/L) according to the local laboratory normal values (40–190 UI/L) and EFNS definition (European Federation of Neurological Societies) [31]. The last were referred to the Neuromuscular Regional Referral Center for further investigations. The parents of the five children who did not attend specialist evaluation were questioned by the PCP and they reported a lack of perception of a clinical problem, a feeling of excessive clinical attention, or fear of test results. Of the ten children with normal CK level and normal neuromuscular examination, two had signs of possible central nervous system involvement (one had hypertonia and the other had signs of neurodevelopmental disorder). Both were sent to the specific diagnostic and therapeutic service. The other eight children are continuing development monitoring at the PCP.

Of the four children sent to the Neuromuscular Regional Referral Center, two presented hyperCKemia (300–700 IU/L) but were negative for the complete neuromuscular exams performed (blood count, liver and kidney blood function tests, ammonium, lactate, electrolytes, urinary organic acids, plasma amino acids, acylcarnitines, very long chain fatty acids, alfa-glucosidase dosage on Dried Blood Spot (DBS), genetic tests for SMA, DMD, and MD1), for which they were referred to an expanded genetic panel (Next Generation Sequencing panel). In two children, the CK level and the neuromuscular examinations were normal as well, but in consideration of the motor delay and the clinical signs (ligamentous hyperlaxity), they were referred to the genetics service for expanded studies in the field of collagenopathies/syndromic conditions.

## 4. Discussion

In this study, a model to speed up the standard organizational hub and spoke model involving primary care pediatricians, a district Child Neuropsychiatry Unit, and a Neuromuscular Regional Referral Center was implemented to reduce the diagnostic delay for NMDs. The model proved to be sustainable because no extra burden was added to PCPs: they had just to note the ‘red flags’ during the Health Status Check visits, which are mandatory in the Italian National Health System. This model could be applied to other areas, wherever there is the availability of specialist child neuropsychiatric services. In other countries, where regular and mandatory assessments of the child health status are not performed, or a national health system is not present, appropriate modifications should be considered.

Considering the known incidence of DMD (1 case per 5000 live male births [32]), SMA (1 case per 7524 live births [19]), and PD (1 case per 50,000 inhabitants [33]) and considering live births in the Piemonte Region (25,937 in 2020, [30]), we would expect 2–3 DMD, 3 SMA, and 0–1 PD new cases per year throughout the Region. As expected, no new diagnoses were made in the ASL TO5 district in one year (about 300,000 inhabitants, out of about 4,300,000 inhabitants in Piemonte).

According to our preliminary project results (percentage of referral for motor milestone delay 0.7% of the total Health Status Check visits, CI 95% 0.3–1) we expected about 70 referred children. However, in the pilot project the children sent for a motor developmental delay (according to the same ‘red flag’ model) had been 20 (0.2% of the total Health Status Check visits, CI 95% 0.1–0.3). Another critical issue that emerged from the application of this model was the high number of dropouts: both parents who did not bring their child to the scheduled specialist visit, and parents who, after the first negative examination (CK), did not want to continue the investigations. It should be noted that four out of five dropouts occurred in the first 9 months of the child’s life, two even in the first month. The difference in the number of referred children and the high number of dropouts suggest that a greater communication effort is needed, perhaps with simple and reassuring graphic material to reduce the parents’ concern for the specialist visit. An organizational model with recall from PCPs after the missed specialistic visit and in which the exams are carried out all together on a single sample could improve the compliance and the completeness of exams. Finally, a single training session is probably not enough to effectively convey information to all health care professionals on treatments’ efficacy, considering the rapidly changing scenario.

From the diagnostic point of view, we had confirmation of the high incidence of hyperCKemia (in this study, 4 out of 20 patients reported for motor developmental delay). As is known, the increase in serum CK level is commonly elevated in patients with skeletal muscle disease or in patient with NMDs with secondary involvement of muscle (e.g., neurogenic disorders) [31], and also in various systemic disorders [34]. In pauci-symptomatic patients with hyperCKemia, as in the case of the four patients in the study, a careful diagnostic study is recommended, both to exclude non-neuromuscular causes of hyperCKemia and to recognize neuromuscular diseases promptly.

Finally, the finding of diagnoses of non-neuromuscular diseases in this study (neurodevelopmental disorders, genetic syndromes, central movement disorders) can be seen as a confounding factor or as a confirmation of the efficacy of a system based on the evaluation of neuropsychomotor development at primary care level for the early identification of diseases in children and the activation of specific diagnostic tests or extended genetic studies such as whole exome or genome sequencing [35,36]. Many rare neurological diseases have early signs that affect motor, language, or cognitive development in children. For some of them, DMTs are currently available, with dramatic effects on survival and quality of life. For others such treatments are expected in the coming years. For some diseases, an NBS program is possible due to the presence of a specific biomarker or a prevalent molecular defect. For others, only costly targeted or extended sequencing of the genome could identify the defect, which is not applicable on a demographic basis. Additionally, for most of them, there are no DMTs available, giving rise to concern about pre-symptomatic identification, while remaining beneficial for the child and parents to receive early diagnosis for standards of care and genetic counseling. Expanding genetic screening programs with this type of “red flag” monitoring program could therefore facilitate the diagnosis of rare neurological diseases.

## 5. Conclusions

In the era of DMTs for metabolic, genetic and neuromuscular diseases, with many therapies expected in the coming years, early diagnosis is becoming an unavoidable public health necessity. Specific newborn screening (NBS) programs are the gold standard to identify individuals with some rare treatable conditions (e.g., SMA), but so far, not all patients with treatable rare diseases are screenable through NBS techniques, and whole-exome and genome sequencing as screening tools for all newborns pose many scientific, economic and ethical concerns.

This pilot study demonstrates the feasibility of conducting a universal screening for rare genetic NMD as part of the routine health visits conducted by PCP in the first two years of life. This model could expand genetic screening programs to intercept and early diagnose neurodevelopmental disorders. However, many questions remain open, especially as regards the real effectiveness of this system and its reproducibility in different health systems.

## Figures and Tables

**Figure 1 children-09-01063-f001:**
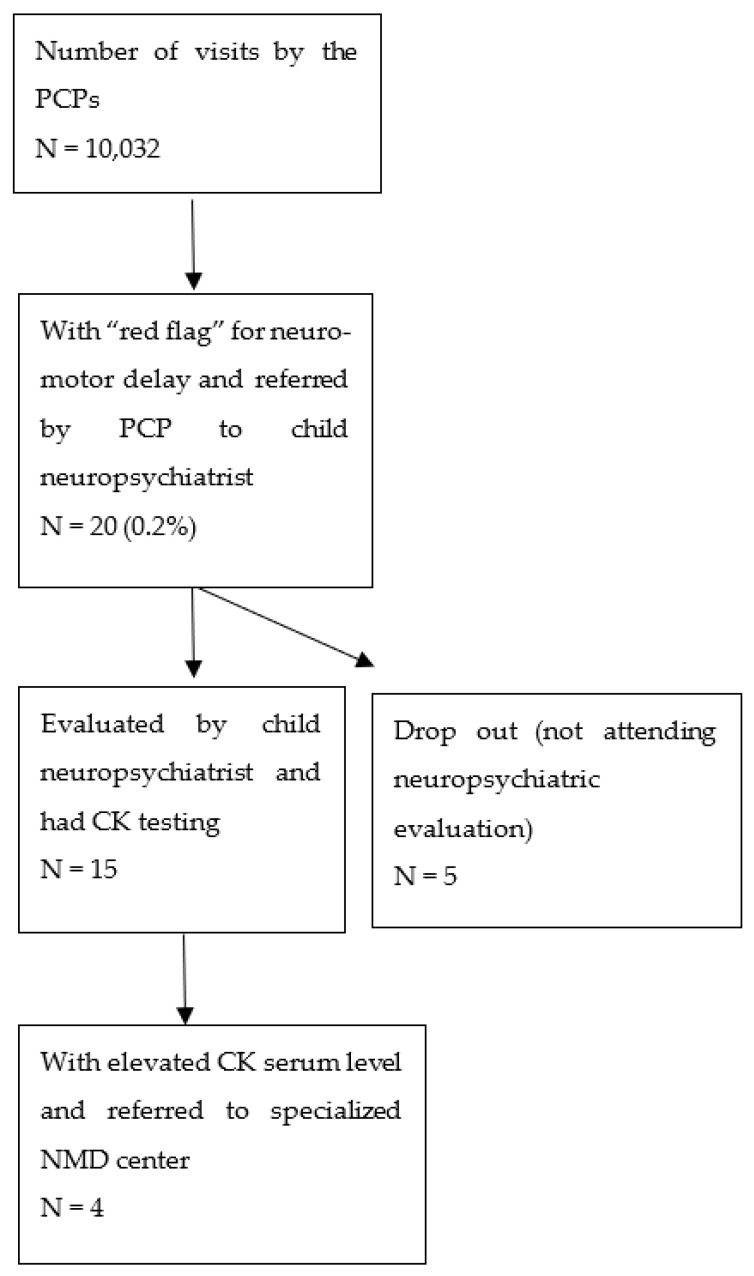
Flow of visits with “red flags” through the screening process.

**Table 1 children-09-01063-t001:** Developmental motor milestones selected from the Health Check Visits charts.

Health Check Visit	Key Question (with Red Flag if the Child Has Not Reached the Specified Milestone)
15 days	Does the child try to lift his/her head?
2–3 months	Has the child good head control?
4–5 months	Does the child point his/her feet?
8 months	Can the child sit?
10–11 months	Does the child crawl or shuffle?
15–18 months	Does the child walk (few steps)?
24–30 months	Is there a language developmental delay?

**Table 2 children-09-01063-t002:** Details of Health Status Check Visits carried out in 2021 at ASL TO5 District.

Health Check Visit	Number of Visits	Number of Children Referred for Red Flag
15 days	1594	2
2–3 months	1570	2
4–5 months	1573	1
8 months	1523	8
10–11 months	909	5
15–18 months	1534	2
24–30 months	1329	0
Total	10,032	

**Table 3 children-09-01063-t003:** Details of referrals according to positive ‘red flags’. **🚩**: red flag for which a child was referred to the neuropsychiatric evaluation at district level. Drop out: the parents and the patient did not attend the neuropsychiatric evaluation at district level. Protocol: diagnostic protocol prescribed by the district neuropsychiatrist for motor development delay, including CK, DBS for Pompe disease, and genetic test for SMA. NM Center: referral to the regional center for clinical signs. CNS: Central Nervous System.

Children	🚩	Results
01	15 days	Drop out
02	15 days	Drop out
03	2–3 months	Protocol (Normal CK)
04	2–3 months	NM Center (HyperCKemia)
05	4–5 months	Protocol (Normal CK)
06	8 months	Drop out
07	8 months	Drop out
08	8 months	Protocol (Normal CK)
09	8 months	Protocol (Normal CK)
10	8 months	Protocol (Normal CK)
11	8 months	Protocol (HyperCKemia, referred to NM Center)
12	8 months	NM Center (HyperCKemia)
13	8 months	NM Center (Normal CK, Genetic)
14	10–11 months	Drop out
15	10–11 months	Protocol (Normal CK, CNS disorder)
16	10–11 months	Protocol (Normal CK, CNS disorder)
17	10–11 months	Protocol (Normal CK)
18	10–11 months	NM Center (Normal CK, Genetic)
19	15–18 months	Protocol (Normal CK)
20	15–18 months	Protocol (HyperCKemia, referred to NM Center)

## Data Availability

Not applicable.

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
