# Peer review of "Improving Recognition of Treatable Rare Neuromuscular Disorders in Primary Care: A Pilot Feasibility Study"

_children, 2022, doi:10.3390/children9071063_

Round 1

Reviewer 1 Report

Early diagnosis of children with neuromuscular and neurodevelopmental disorders is a very important task; it always has psychosocial and, increasingly, therapeutic consequences.

The authors present the results of a feasibility study of a complex early diagnosis system for neuromuscular disorders in infancy and early childhood, based on a three-tiered detection and diagnosis system starting at the primary care pediatrician (PCP) level with five age-related items on motor development, leading to a local neuropsychiatric specialist at the second tier, and finally to a regional specialty center at the third tier with laboratory tests for the three major disorders.

Of 10 032 children tested at the PCP level, 20 progressed to the second level, and after exclusion of several cases with alternative CNS or connective tissue disorders, 4 children were found to have moderately elevated CK at the third level (although a definitive etiologic diagnosis has not yet been established).

This manuscript highlights the possibilities and limitations of a clinically based screening program in very rare diseases and should be published to share the experience with the interested public.

However, I would like to add some thoughts that could be included in the discussion (to the extent that they have not already been addressed by the authors):

1. I believe that the three-step diagnostic pathway from primary care physicians to local specialists to regional centers with special expertise is the usual way to find a diagnosis in developed countries; the task is only to speed up this process.

2. All NMDs are rare or extremely rare diseases. In 10 000 children, one would expect only 1-2 patients with DMD and SMA (they were not found in this study!), and the other diagnoses are 10-100-times rarer. Although I am not an epidemiologist, I believe that a subsequent study to validate this screening tool would need to include more than 100,000 children.

3. PCPs identified only 20 cases instead of the expected 70. I do not believe this is a problem due to lack of knowledge. I am sure that the Italian PCPs are well-trained pediatricians and that they have ample day-to-day experience with normal, borderline, and abnormally developed children. But they also know from daily experience that the suspicion of a developmental problem is a very heavy burden for the families, which should be imposed on them only when the suspicion is really high. Thus, they are accustomed to "wait and see" further development for a while before informing families of the need for further diagnostics. Similar backgrounds are certainly the cause of the high drop-out rate due to the refusal of families for further diagnostics. This should be taken into account when organizing such screening programs.

4. Probably PCP screening could be improved by asking for similar cases in the family history, since all these NMDs are hereditary diseases.

5. Hypotonia and delayed motor milestones in the first year of life are very non-specific "red flags" that are much more likely to indicate normal variants (5% of children never crawl), mental retardation, and CNS disorders than the very rare NMDs. This study has also shown this. Although I know that this screening project was initiated by a clinical research group focused on NMDs, I think it would make much more sense to expand the target at the expert level to all neurodevelopmental disorders and count all of these diagnoses as "positive" outcomes. This would also greatly increase the success rate of this feasibility study.

6 The highest diagnostic time pressure to date is in infantile SMA, as it is increasingly clear that available gene therapy is most successful when initiated before the onset of the first symptoms. Therefore, in SMA, molecular genetic newborn screening is the only way to make this diagnosis in a timely manner and cannot be replaced by clinically oriented screening.

7. Considering the fact that this is only a feasibility study with many problems still to be discussed, I think the introduction with the detailed description of therapeutic developments is too long. In citation [4}, the number of neuromuscular or neurological diseases with AAV-based treatment studies should be given.

Author Response

  1. I believe that the three-step diagnostic pathway from primary care physicians to local specialists to regional centers with special expertise is the usual way to find a diagnosis in developed countries; the task is only to speed up this process.

We fully agree with the reviewer: the three-step diagnostic pathway from PCPs to local specialists to regional centers is the usual way to find a diagnosis in developed countries. The need is to speed up the process and, for this reason, the proposal we tested included the indication for selected diagnostic tests (CK, SMA and PD) at the second level (local neuropsychiatric service) for all children with red flags, reserving cases with clear signs of neuromuscular involvement the extended diagnostic protocol from the third level (regional NM center). As commented, we have found some pitfalls of this model, including the non-execution of the proposed exams.

We have better clarified the concept of speeding up the process (line 175, we added “quick”; lines 196-197, we added time for the first neuropsychiatric assessment according to our proposed protocol; lines 265-268, we modified the sentence to “In this study, a model to speed up the standard organizational hub and spoke model involving primary care pediatricians, a district Child Neuropsychiatry Unit, and a Neuromuscular Regional Referral Center, is proposed to reduce the diagnostic delay for neuromuscular diseases”)

  1. All NMDs are rare or extremely rare diseases. In 10 000 children, one would expect only 1-2 patients with DMD and SMA (they were not found in this study!), and the other diagnoses are 10-100-times rarer. Although I am not an epidemiologist, I believe that a subsequent study to validate this screening tool would need to include more than 100,000 children.

We would like to clarify that we have reported the number of Health Status Check visits (10,032 visits) and not of children evaluated, because the regional system acquires these data. Each child was evaluated more than once during our pilot year. Since the number of visits for each child depends on the age of the child, we cannot derive the exact number of children, but it is far less than 10,000. This explain why we have not found any SMA/DMD diagnosis.

We completely agree that the number of children would have to be much higher to reach potency for an epidemiological and efficacy study. Following this feasibility project, and probably with some correction of the pitfalls we encountered, our idea is to extend the study to the whole of our region.

  1. PCPs identified only 20 cases instead of the expected 70. I do not believe this is a problem due to lack of knowledge. I am sure that the Italian PCPs are well-trained pediatricians and that they have ample day-to-day experience with normal, borderline, and abnormally developed children. But they also know from daily experience that the suspicion of a developmental problem is a very heavy burden for the families, which should be imposed on them only when the suspicion is really high. Thus, they are accustomed to "wait and see" further development for a while before informing families of the need for further diagnostics. Similar backgrounds are certainly the cause of the high drop-out rate due to the refusal of families for further diagnostics. This should be taken into account when organizing such screening programs.

We totally agree with this comment and thank the reviewer because we probably misstated the concept: Italian PCPs are really well trained in detecting signs of abnormal development in children. What we meant by lack of knowledge was referring to the new DMTs for rare diseases, as this is a rapidly changing scenario. The high dropout rate due to families' refusal for further diagnosis could be a problem with screening based on early clinical and nonspecific signs, as our study has shown. Probably the motivation of health professionals (all), based on knowledge of the advantage of early treatment, could lead to the structuring of more efficient mechanisms for verifying compliance and to a greater communicative impact in direct interaction with parents.

We clarified this concept modifying sentences from line 281 to line 303.

  1. Probably PCP screening could be improved by asking for similar cases in the family history, since all these NMDs are hereditary diseases.

We appreciate this proposal. In the development phase of the program we had thought of expanding the information captured by the PCPs (collection of anamnestic information or verification of clinical signs such as tendon reflexes) but in the end the whole group decided to be as simple as possible, at the cost of a low specificity. In fact, as the reviewer suggests, anamnestic assessment of family history could further improve sensitivity, and we might consider including this point in the regional extension of the project.

  1. Hypotonia and delayed motor milestones in the first year of life are very non-specific "red flags" that are much more likely to indicate normal variants (5% of children never crawl), mental retardation, and CNS disorders than the very rare NMDs. This study has also shown this. Although I know that this screening project was initiated by a clinical research group focused on NMDs, I think it would make much more sense to expand the target at the expert level to all neurodevelopmental disorders and count all of these diagnoses as "positive" outcomes. This would also greatly increase the success rate of this feasibility study.

We are very grateful for this comment. As correctly pointed out by the reviewer, we are a group focused on NMDs, but we think this model can be extended to all neurodevelopmental disorders. With the anticipated arrival of other DMTs and the evolution of multispecialistic standards of care, and the possibility to sequence genome / exome, early detection of many rare diseases is becoming a health planning priority. Almost all neurodevelopmental disorders initially manifest as nonspecific changes in developmental trajectories, so interception of these signs of abnormal development could be a common strategy for all of these disorders. The idea of using this model (as well as the one already used for Autism Spectrum Disorders) for all neurodevelopmental disorders is rapidly spreading.

We modified the sentences from line 326 to 332 to expand this concept.

6 The highest diagnostic time pressure to date is in infantile SMA, as it is increasingly clear that available gene therapy is most successful when initiated before the onset of the first symptoms. Therefore, in SMA, molecular genetic newborn screening is the only way to make this diagnosis in a timely manner and cannot be replaced by clinically oriented screening.

When we ran this pilot project, NBS for SMA was not available in our region. We are implementing it now. We agree with the reviewer that for SMA the most correct model is NBS (although the small percentage of patients with point mutations are not identified by this method). The model we propose should therefore evolve for diseases for which NBS programs are not yet implemented or are indeed infeasible.

We tried to clarify this concept from line 321 to 326.

  1. Considering the fact that this is only a feasibility study with many problems still to be discussed, I think the introduction with the detailed description of therapeutic developments is too long.

As suggested, we refined the details of the introduction.

In citation [4}, the number of neuromuscular or neurological diseases with AAV-based treatment studies should be given.

The absolute number over the 149 selected trials is not obtainable from the supplementary materials of the paper, but we cited the increase in clinical trials for both CNS and musculoskeletal disorders reported.

Reviewer 2 Report

This is an interesting and timely paper. The data presented, on a pilot study that demonstrates the feasibility of an early detection program for a collection of NMDs is of real importance and interest to the research and clinical communities associated each disorder. For long scale detection, with multiple disorders, using normal / combined genetic screening analysis (like NBS blood spot analysis) would be extremely expensive- therefore, this type of early detection through milestone analysis could be if interest. However, the issue (especially with disorders like SMA, where there is a genetic test and three separate treatments), is that relying on milestones (like currently happens in the UK) means treatment only happens after clinical symptoms emerge, which means the MN targets have been reduced and the treatment will not be efffective.

This means the main weakness with the paper is that it does not effectively review how this type of "red-flag" monitoring program could be implemented alongside a genetic screening program. The discussion should be expanded to review what this proposed approach offers, how it expands current approaches and the benefits of running them (potentially) alongside each other. 

Author Response

We thank the reviewer for this comment. It highlights the possibility of expanding current diagnostic approaches (diagnosis based on traditional clinical signs and subsequent investigations, newborn genetic screening) with different programs to optimize the early diagnosis of diseases with different clinical courses, different diagnostic possibilities and different treatments.

We have expanded the discussion including the reviewer’s suggestions.